# The Role of In Vitro Immune Response Assessment for Biomaterials

**DOI:** 10.3390/jfb10030031

**Published:** 2019-07-12

**Authors:** Alistair Lock, Jillian Cornish, David S. Musson

**Affiliations:** Department of Medicine, University of Auckland, Auckland 1142, New Zealand

**Keywords:** in vitro, immune response, immunogenicity, biomaterials, tissue regeneration

## Abstract

Grafts are required to restore tissue integrity and function. However, current gold standard autografting techniques yield limited harvest, with high rates of complication. In the search for viable substitutes, the number of biomaterials being developed and studied has increased rapidly. To date, low clinical uptake has accompanied inherently high failure rates, with immune rejection a specific and common end result. The objective of this review article was to evaluate published immune assays evaluating biomaterials, and to stress the value that incorporating immune assessment into evaluations carries. Immunogenicity assays have had three areas of focus: cell viability, maturation and activation, with the latter being the focus in the majority of the literature due to its relevance to functional outcomes. With recent studies suggesting poor correlation between current in vitro and in vivo testing of biomaterials, in vitro immune response assays may be more relevant and enhance ability in predicting acceptance prior to in vivo application. Uptake of in vitro immune response assessment will allow for substantial reductions in experimental time and resources, including unnecessary and unethical animal use, with a simultaneous decrease in inappropriate biomaterials reaching clinic. This improvement in bench to bedside safety is paramount to reduce patient harm.

## 1. Introduction

Grafting is required to improve healing and restore tissue integrity and is used in cases where the body’s reparative processes are impaired or a defect size exceeds that of the body’s healing ability [1]. The current gold standard graft material is autograft, using the patient’s own tissue to enhance healing. However, due to the increased surgical time, donor site morbidity and limited harvest of some tissues, alternative options are sought [2]. In the search for viable substitutes, the number of biomaterials being developed has rapidly increased and contributes a large part of an estimated $400B (USD) worldwide medical device market [3,4].

Currently, clinical uptake of biomaterials is poor. Using orthopaedic surgery as an example, there were 47 biomaterials approved for human use in bone regeneration as of 2016. However, no single material is currently preferred over autograft, with the highest reported use rates approximating 25% [5]. This low use rate for biomaterials is due to their inherently high failure rates, which translates to a lack of confidence from the surgeons who would use them. 

In 2017, the United States food and drug administration’s (FDA) manufacturer and user facility device experience database (MAUDE) reported 944,980 adverse events involving medical devices, with event types ranging from mild to severe [6]. Although this number seems large, the true burden is likely worse, as reporting is voluntary and many adverse clinical outcomes can be missed.

Immune rejection has been a common end result of the clinical use of prevailing biomaterials. One example is the collagen-based RESTORE™ implant, which was used to enhance soft tissue healing in rotator cuff repair surgery. After four of the initial 19 clinical cases resulted in severe inflammatory reaction, requiring reoperation and implant removal, it was recalled [7]. In other cases, immune rejection reaction has led to more severe patient outcomes. For example, the off-label use of the InFUSE™ bone scaffold in cervical fusion has caused cases of acute airway compression due to the soft tissue swelling, a characteristic of immune rejection [8]. Thirty-eight cases of severe dysphagia and respiratory failure causing mortality triggered the FDA to release a public health notification for this material in 2008 [9]. 

## 2. Current In Vitro Testing of Biomaterials

The standard in vitro testing protocols for assessing novel biomaterials generally follow the International Organisation of Standardisation (ISO) protocols for the biological evaluation of medical devices [10]. These focus on cytotoxicity and are general across all medical device types. 

Specific purpose in vitro testing is also often carried out by research laboratories, using a variety of immortalised cell lines, primary cells and stem cells relevant to the target tissue. Here, the focus is on assessing cytocompatibility. The ability of cells to grow on the materials using metabolic assays or qualitative imaging or the ability of cells to differentiate on the materials using gene expression and protein deposition assays are commonly studied for this purpose [4].

However, a recent European multicentre analysis of biomaterial assessments showed a surprisingly poor correlation between current in vitro testing of biomaterials and in vivo outcomes for bone regeneration [11]. This study concluded that in vitro evaluations of biomaterials are not suitable to predict in vivo acceptance and highlights the role for improving current in vitro assessment protocols and developing more relevant in vitro assays.

Given the importance of the immune response to foreign materials, and its seemingly pivotal role in the clinical outcomes for novel biomaterials, it is surprising that current evaluation protocols do not incorporate immune testing into the assessment criteria from an early stage. Clearly, there is a substantial role for greater understanding about how novel biomaterials interact with the immune system to improve patient safety. Elements of the foreign body response can be evaluated to predict immune response early and determine likelihood of acceptance, which would negotiate undesirable local or systemic responses at the clinical level. 

## 3. Foreign Body Response to Implanted Biomaterials

Based on previous work, it is clear that it is important to understand the foreign body response to implanted biomaterials. The foreign body response (summarized in Figure 1) is initiated following introduction of a biomaterial into a living host. This causes the release of activational cytokines and commences a chain of events, starting with protein adsorption [12]. Host plasma proteins adhere to the biomaterial surface and act as further activation signals for the onset of the inflammatory response. Following protein adsorption, the processes of acute and chronic inflammation occur [12].

The duration ofacute inflammation ranges from minutes to days and is defined by polymorphonuclear leucocyte presence at the site of repair [12,13]. This is mainly neutrophil driven; however, eosinophils and basophils are also present. The goal of this phase is to activate host mechanisms and clear debris from surgical injury [12]. 

Chronic inflammation starts after the acute phase and is defined by mononuclear leukocyte presence [12,14]. Macrophages and other monocytes play a central and critical role in early wound healing in the *innate* phase, releasing enzymes important in tissue reorganisation and angiogenesis [12]. Monocytes also partake in phagocytosis of debris and release of cytokines and growth factors which stimulate the migration and proliferation of other cell types, including fibroblasts and mesenchymal stem cells [12]. This is significant as these cells further stimulate monocytes in a positive feedback cycle. Lymphocytes are also mononuclear leukocytes which are involved later in the *acquired* phase of chronic healing [15]. 

Several studies to date have managed to incorporate immune testing into their biomaterial evaluations. The current literature will be discussed, highlighting the gap in the literature to date and the role for assessments of in vitro immune response to enable predictions of in vivo application.

## 4. In Vitro Immunogenic Evaluation

Cell types chosen to evaluate immunogenic response include either primary cells of the chronic *innate* immune response, such as peripheral blood mononuclear cells (PBMCs), peritoneal and splenic macrophages or an established acceptable cell line model. Other *innate* immune cell types such as dendritic cells and natural killer (NK) cells have also selectively been used as primary cells, with *acquired* immune response cells such as T and B lymphocytes used far less often. 

Immunogenic evaluations have generally had three areas of focus: cellular viability, maturation and activation, with the latter being most prominent in the literature.

### 4.1. Viability Studies

Measurements of viability are one way of determining immune cell cytotoxicity in response to a material, suggesting that if cells respond negatively and viability reduces, that biomaterial might not support optimal wound healing in vivo. Conversely, seeing large increases in immune cell viability may also not be the optimal outcome, as it suggests some level of activation in response to the material. Therefore, using immune cell viability as a single assay for determining immunogenicity may not be the best method for determining in vivo acceptance.

Figure 2 discusses the pros and cons of various viability assays. Visualising the morphology and number of immune cells that populate the surface of a biomaterial is one common method for assessing cell viability. Cells can be imaged directly on a material using microscopy and staining protocols such as calcein-acetoxymethyl (AM) live-dead [16] and actin and nuclei staining [17]. This provides a relatively simple and quick visualisation of immune cell response to a given material. 

Measuring the viability of chosen cells interacting with biomaterials through metabolic assays, such as the alamarBlue^®^ and glucose-6-phosphate dehydrogenase (G6PD) assays are often used to assess viability of cells of the innate immune response such as PBMCs, macrophages, dendritic cells or an established cell line model [18,19].

Cell viability assessment for cells of the *acquired* immune response, such as T and B lymphocytes, have been very limited. Various B and T lymphocyte proliferation, cytotoxicity and antibody-production assays have been used, but these appear to have been limited to electrospun polydioxanone-elastin biomaterial blends produced by one group [20,21].

### 4.2. Maturation Studies

Studies of immune cell maturation have largely focussed on assessing cell surface markers. Cell surface markers are produced at varying stages during immune cell maturation, so measuring these in cells that have been exposed to a biomaterial allows researchers and clinicians to determine if a given material is inducing a cell to differentiate/mature.

Markers for evaluation are chosen based on markers of pro- and anti-inflammation. These come from two known functional sub-types originally described in macrophages: M1 (classically activated) and M2 (alternatively activated) [4,12,15,22,23]. The M1 lineage functions in pro-local and systemic inflammation, and if persistent, can result in rejection of biomaterial [4]. The M2 sub-type has three independent types: M2a (allergic responses), M2b (immune regulation) and M2c (wound-healing) [4,12,15,22,23]. Although important, the M2a and M2b types serve lesser roles in biocompatibility evaluations. An M2c heavy response is known to have anti-inflammatory effects with promotion of wound healing, extracellular matrix deposition and tissue remodelling [22,23]. 

Examples of cellular surface markers chosen in the literature include expression of CD86/CCR7 for pro-inflammation/M1 lineage and CD163/CD206 for anti-inflammation/M2 lineage, respectively [12].

Immunohistochemistry is one method used by researchers to investigate cell surface marker expression and localisation. Fearing and Van Dyke (2014) [24] used this technique to assess immunogenic response to a keratin biomaterial by staining for two specific markers of maturation using a THP-1 cell line. These were CD86, a protein expressed on antigen presenting cells which provides signals for T cell activation and survival, and CD206, a M2 subtype pattern recognition receptor which mediates phagocytosis and promotes healing. The group used confocal microscopy to quantitatively search for THP-1 cells labelled as CD86+/CD206− (pro-inflammatory) and CD86-/CD206+ (anti-inflammatory), showing that exposure of the THP-1 cell line to keratin biomaterial substrates causes a bias towards a favourable M2 phenotype [24]. 

Secondly, immune cell surface markers have been investigated using fluorescence activated cell sorting (FACS), which is a particular form of flow cytometry which enables a mixture of cells to be sorted according to cell surface markers tagged with fluorescent labels to allow quantification [25]. Park and Babensee (2012) used this technique to demonstrate the range of dendritic cell surface marker phenotypes induced by biomaterials. The group observed for the presence/absence of CD32, CD40, CD44, CD80, CD83, CD86, CD206, HLA-DQ, HLA-DR and Annexin V cell surface markers, with phenotypes used to provide selection criteria for biomaterials used in immunomodulatory applications [26]. Musson et al. (2015) similarly examined cell surface marker expression of immature dendritic cells exposed to a novel kitted, non-mulberry silk fibroin biomaterial (Spidrex^®^) with potential applications in tendon regeneration. Here they showed similar levels of mature dendritic cell surface markers CD80, CD83 and CD86 in cells exposed to the silk biomaterial and cells exposed to commonly used surgical suture material, but lower levels than those in cells exposed to LPS [27]. The authors, therefore, suggested that the biomaterial would likely have a similar, acceptable in vivo response to the suture material.

### 4.3. Activation Studies

Measuring activation of chosen immune cells at selected time-frames has been the most common method of characterising in vitro immunogenic response to foreign biomaterials. The reason for the predominance of activational studies in the literature is likely due to their ability to look at a wider selection of both pro-inflammatory (M1) and anti-inflammatory (M2) markers, compared with maturation studies. This wider selection offers more information for researchers to allow them to make predictions for functional outcome. For example, if there was a high pro- to anti-inflammatory ratio, the material would likely induce a severe foreign body response, while high anti- to pro-inflammatory ratios have been associated with positive healing outcomes in vivo [28,29].

Table 1 summaries common cytokines used to assess immune rejection or acceptance for biomaterials. Pro-inflammation can be characterised by the release of activational cytokines such as IL-1α, IL-1β, IL-6, IL-8 and TNFα. This can also be measured functionally by examining elements such as monocytic phagocytosis, and nitric oxide and reactive oxygen species (ROS) production. Anti-inflammation consistent with wound healing can be characterised by the release of anti-inflammatory cytokines such as IL-1RA, IL-4, IL-10, IL-13 and TGFβ [22,23]. 

Studies of activation often rely on comparisons with materials that are known to positively activate immune cells to provide some clinical context. Examples of these are lipopolysaccharides (LPS) [27] or clinically used suture materials [30,31]. 

### 4.4. Protein-Based Assays

One method of measuring inflammatory activational markers released from immune cells is enzyme-linked immunosorbent assay (ELISA). An ELISA is a plate-based assay technique which can quantify specific proteins from a liquid sample using highly specific antibodies, which are linked to an enzyme. Detection is accomplished by assessing conjugated enzyme activity via a colour change [32]. This assay has been used to quantify the concentration of inflammatory cytokine, chemokine and growth factors released from immune cells exposed to novel materials in many studies, ranging from those measuring only a few inflammatory markers, to those measuring several. The pro-inflammatory cytokines that have been assessed most often in single-marker assays are IL-1β, TNFα and IL-6 [26,33,34,35,36].

The more recent ELISA-based studies have tended to use multiple kits to measure several markers of immune cell activation. The benefit of this is it provides simultaneous information on secreted levels of proteins that are associated with both pro- and anti-inflammatory properties. Early unbalanced immune reactions with prolonged pro-inflammation has been shown to delay wound healing responses in both cutaneous and bony wound models [37,38,39]. Hence, the information gained from these multiple markers can allow one to make predictions on in vivo immune acceptance for a given biomaterial. Garg et al. (2013) assessed TNFα, IL-6, VEGF (vascular endothelial growth factor), TGFβ1, FGF (fibroblast growth factor), MIP1α and MCP-1 production from murine bone-marrow-derived macrophages, in an evaluation of electrospun polydioxanone biomaterials with varying polymer concentrations and fibre/pore size, showing that higher concentrations of polymer polarised macrophages toward an M2 phenotype, with a downregulation of M1 markers [40]. Alternatively, Almeida et al. (2014) chose to measure secretion of TNFα, IL-6, IL-10, TGFβ1 and the IL-12/IL-23 p40 subunit, when assessing 3D printed, biodegradable PLA-, PLA/calcium phosphate and chitosan-based biomaterials using PBMCs. The group showed that chitosan-based biomaterials elicited increased secretion of TNFα, an M1 marker, while PLA-based scaffolds induced higher production of IL-10, a M2 marker, indicating PLA may be a more compatible material for them to use in future assessments [17]. 

However, purchasing of multiple kits with separate assays per individual marker can be both time and labour intensive. To alleviate this issue, multi-analyte ELISA kits have been developed. In their evaluation of a keratin biomaterial using a THP-1 cell line, Fearing and Van Dyke (2014) detected eleven different inflammatory markers simultaneously using a panel of capture antibodies for IL-1α, IL-1β, IL-2, IL-4, IL-6, IL-10, IL-12, IL-17A, IFNγ (interferon gamma), TNFα and GM-CSF (granulocyte-macrophage colony-stimulating factor) [24]. They showed that exposure of this monolytic cell line to keratin biomaterial substrates causes a bias towards an M2 phenotype, influencing a greater production of anti-inflammatory cytokines, with decreased amounts of pro-inflammatory cytokines. This work indicated that the use of a keratin biomaterial may regulate a positive remodelling response in regenerative medicine [24].

Another way to simultaneously study immune markers of activation from cell media is with protein microarrays. Protein microarray can qualitatively measure a large number of cytokines, chemokines and growth factors [41]. Chang et al. (2008) assessed five peripheral blood lymphocyte (PBL) and PBMC-seeded hydrophobic, hydrophilic and ionic polymers by determining the presence/absence of 79 markers in the cell-culture supernatant [42]. They demonstrated that hydrophilic/neutral and hydrophilic/anionic polymers increased IL-8 and TNFα production relative to hydrophilic/cationic and hydrophobic polymers, indicating that hydrophilic/neutral and hydrophilic/anionic substrates may push macrophages towards an unfavourable pro-inflammatory M1 phenotype during the foreign body reaction. Hydrophilic/cationic surfaces also induced relatively high levels of IL-10 production, indicating hydrophilic/cationic surfaces induce a favourable M2 driven anti-inflammatory response. 

The problem with the protein microarray, however, is that the signal produced is qualitative. To quantitatively measure detected marker concentrations from the large set available, validation using ELISA-based assays is often required.

A final method which can concurrently and quantitatively detect multiple markers of immune cell activation is the cytometric bead array (CBA). Most forms of CBA can quantitatively measure up to 30 proteins (cytokines, chemokines and growth factors) from cell supernatant, using antibody-coated beads to efficiently capture analytes. Each bead has unique fluorescence which can be detected using flow cytometry [43]. Cytometric bead array assays have been performed on a range of immune cell types, with again, varying activational markers selected for assessment. Grotenhuis et al. (2014) utilised PBMCs in an in vitro model studying the foreign body response to polypropylene, collagen and polyethylene terephthalate-based biomaterials. Supernatant protein levels of IL-1β, IL-6, TNFα, MCP3, MIP-1α, IL-1RA, CCL5 (chemokine ligand-5) and CCL22 (chemokine ligand-22) were determined using this multiplex system, with findings suggesting secretion in a biomaterial-dependent manner [44]. Musson et al. (2015) also used CBAs to simultaneously measured cytokine concentrations from for IL-10, IL-12, IL-1β, IL-6 and TNFα in dendritic cells exposed to a silk biomaterial, showing enhancement of activation markers in a similar fashion to a commonly used surgical suture material, but less than LPS [27]. The information obtained in this study suggested that this material may induce an immune response similar to that of clinically accepted suture material.

### 4.5. Relative Gene Expression-Based Assays

Activational studies can also focus on gene expression of inflammatory markers from immune cells interacting with respective biomaterials. The concept provides simultaneous information on relative gene expression for multiple genes associated with both pro- and anti-inflammatory properties, allowing one to make predictions on in vivo immune acceptance for a biomaterial. One method of doing this is semi-quantitative reverse transcription polymerase chain reaction (semi-quantitative RT-PCR). This method semi-quantitatively detects levels of target gene expression, using a messenger RNA (mRNA) template obtained from chosen cells, and gel electrophoresis. Brodbeck et al. (2002) [45] used this technique in an evaluation of a range of PBMC-seeded hydrophobic, hydrophilic and ionic polymers. Gene expression of *IL-1β*, *IL1RA*, *TNFα*, *IL-6*, *IL-8* and *IL-10* were semi-quantified with comparison to a β-actin housekeeping gene [45]. Their cell culture revealed that *IL-10* expression significantly increased in cells adherent to the hydrophilic/anionic surfaces, but significantly decreased in those adhered to cationic surfaces. These results, therefore, suggest that hydrophilic and anionic surfaces promote an anti-inflammatory type of response, whilst cationic surfaces promote pro-inflammation. Interestingly, this is the opposite finding to that previously described by Chang et al. 2008 [42], indicating that charge is just one aspect influencing the foreign body reaction and other elements such as porosity and material choice also play vital roles.

While RT-PCR allowed for the measurement of relative gene expression levels of pro- and anti-inflammatory cytokines, this technique is now rather outdated and has been replaced by a more modern method of gene expression analysis: quantitative real-time PCR (qPCR). Quantitative real-time PCR enables quantification of amplified target DNA from a sample in real-time, using an oligonucleotide probe designed specifically to anneal to the target gene sequence. During amplification, the probe is cleaved releasing a specific fluorescent marker. Over many cycles, gene-specific fluorescence is measured constantly in real-time, and is directly proportional to the starting amount of the respective target gene in the template sample. Target gene expression is calculated relative to a negative control baseline sample [46]. This real-time analysis shortens experimental time compared with RT-PCR, which in turn increases efficiency of biomaterial assessment. 

Some of the literature utilising qPCR measure only a select few markers of immune cell activation [47]. For example, Jin et al. (2018) showed that THP-1 macrophages produce lower amounts of pro-inflammatory *TNFα* after culture with higher concentrations of magnesium alloys, hence creating interest in the use of biodegradable Mg-based alloys in bone fixation devices or vascular stents [48]. Romero–Gavilan et al. (2019) also showed that incorporating strontium coatings into sol-gel biomaterials increases *TGFβ* expression from a murine macrophage cell line, whilst decreasing TNFα expression. This work indicates that incorporating strontium coatings into biomaterial improves their anti-inflammatory potential [49].

However, an advantage of this technique is the ability to analyse a large variety of markers in real-time, which provides considerable knowledge of immune cell activation with interpretation weighted on M1/M2 subtype ratios. For example, Fottichia et al. (2018) exposed THP-1 cells to various electrospun polycaprolactone substrates and measured the expression of a selection of both pro- and anti-inflammatory genes (*CCR7*, *CD163*, *IL-1β*, *IL-8*, *IL-12*, *TNFα* and *TGFβ1*). Overall, the group observed limited pro-inflammatory response, much lower than that of clinically used suture material, indicating the substrates would likely be well tolerated in vivo [31]. Moreover, different immune activation markers (*IL-10*, *IL-1RA*, *TNFα*, *IL-6*, *IL-1β*, *IFNγ*, *VEGF* and *TGFβ1*) were selected by Chen et al. (2014), when they assessed β-tricalcium-phosphate (*β*-TCP) coated magnesium biomaterials using human bone-marrow-derived macrophages. The group showed that macrophages switched to the M2 phenotype in response to the β-TCP coating, suggesting that different coatings on biomaterials can modulate the scaffold’s osteoimmunomodulatory properties, and allow a researcher to shift the immune environment towards one favouring healing [50]. Another example comes from Lock et al. (2017), who evaluated a selection of commonly used surgical suture materials using the THP-1 cell line, with the intent of improving surgical decision making and patient safety. Six pro- and anti-inflammatory cytokine markers were chosen (*IL-1α*, *IL-1β*, *TNFα*, *IL-8*, *TGFβ1* and *IL-1RA*), with the group suggesting that four of their suture materials cause upregulation of pro-inflammatory markers indicative of an early foreign body reaction, with no balancing from anti-inflammatory markers [30]. Surface polarity was again shown to alter foreign body immune response as Hotchkiss et al. (2019) showed that hydrophilic dental implants expressed higher levels of anti-inflammatory cytokines *IL4*, *IL10* and *TGFβ*, with less expression of proinflammatory cytokines *IL1β*, *IL6*, *IL17A*, *CXCL10* (C-X-C motif chemokine 10) and *TNFα*, compared to hydrophobic implants [51].

Gene expression of cell surface maturation markers can also be quantified using qPCR. Van Putten et al. (2013) quantified expression of both inflammatory cytokines and cell-surface markers from murine bone-marrow-derived macrophages in their evaluation of hexamethylenediisocyanate cross-linked dermal sheep collagen (HDSC) disks. Expression levels of *IL-1β*, *IL-12*, *TNFα*, *IL-10*, *CCR7*, *CD80*, *CD86*, *CD206*, *CD163*, *iNOS* (*inducible nitric oxide synthase*), *Clec10a* (c-type lectin domain family 10 member A), *Ecad* (epithelial cadherin), *Stab1* (*stabilin-1*) and *MSR1* (macrophage scavenger receptor 1) were quantified, with results showing balanced expression between M1 and M2 markers, indicating that HDSC does not heavily favour pro- or anti-wound healing [52]. More recently, Chu et al. (2019) evaluated the immunomodulatory effect of an epigallocatechin-3-gallate (EGCG)-modified membrane. Using qPCR, they showed that collagen discs coated with EGCG downregulated inflammatory markers such as *IL-17RA*, *MMP-1* and *MMP-9*, and upregulated anti-inflammatory markers such as *IL-4*, *IL-10*, *TGFβ*, *CD163* and *CD206*, compared to non-coated collagen controls [53]. Their work implicated positive use of EGCG in both bone and tissue remodelling.

### 4.6. Functional Assays

Studies of immune cell activation also include those assessing monocytic phagocytosis, and nitric oxide and reactive oxygen species (ROS) production. Monocytes phagocytose debris at the repair site in the foreign body response, and based on this, Zenni et al. (1994) used an acid phosphatase assay on peritoneal macrophages cultured with two polyethylene terephthalate biomaterials [54]. However, a more recent technique includes the Cytoselect™ 96 well phagocytosis assay using immunoglobulin G (IgG) opsonised sheep red blood cells used by Smith et al. [20,21]. Increased phagocytic ability of peritoneal macrophages exposed to collagen–chitosan biomaterials has also been demonstrated, compared to controls, by uptake of fluorescently labelled nanoparticles [55].

Nitric oxide (NO–) is a reactive radical produced by monocytes as a defence mechanism against pathogenic attack. Reactive oxygen species, such as O_2_^−^, H_2_O_2_ and OH^−^, are also produced by monocytes in response to certain cytokines or bacterial products [56]. NO– and ROS levels can be quantified from immune cell populations in contact with studied biomaterials to provide information on cellular activation. This can be compared against standards or with other biomaterials to provide comparison. Smith et al. (2009, 2010) quantified total nitrite concentration to measure NO– production from murine spleen-derived macrophages in their evaluation of a selection of electrospun polydioxanone or elastin biomaterial blends. The group found no differences in NO– production between murine spleen-derived macrophages exposed to polydioxanone or elastin [20,21].

Chlopek et al. (2006) used chemiluminescence to assess ROS production from murine peritoneal macrophages cultured with carbon nanotubes. Culture with the carbon nanotubes induced free radical production to a similar level to another biomaterial currently licensed for use in humans, confirming favourable biocompatibility for the research group [57]. Chemiluminescence signal was also used by the previously mentioned Smith et al. (2009, 2010), utilising luminol to generate the signal [20,21]. Although the group demonstrated no differences in NO– production between polydioxanone and elastin blends, they did see a difference in ROS production. Decreased ROS production was seen in macrophages exposed to blends with higher concentrations of elastin, and increased ROS production was seen in macrophages exposed to higher concentrations of polydioxanone, indicating these blends may be more favourable for future assessments.

## 5. Discussion

Given the seemingly pivotal role that immune responses play in the clinical outcomes for novel biomaterials, this review reiterates the important notion that in vitro immune response assessment can offer to improve patient safety. As the FDA’s manufacturer and user facility device experience (MAUDE) database often receives increasing reports of adverse events associated with implanted biomaterials each year [6], new methods of in vitro immune response analysis are required to provide adequate pre-clinical information for the safe use of novel biomaterials in humans.

To date, techniques used to evaluate biomaterials in vitro have been limited in their ability to predict in vivo acceptance [11]. Assessments to date have focused heavily on the concepts of non-immune cell cytotoxicity and cytocompatibility. There has been limited assessments of in vitro immune response.

In vivo studies assessing immune response to a variety of biomaterials have been performed, using a variety of timeframes and techniques for analysis [58]. The ISO Standard 10993-2 specifies the minimum requirements to be satisfied to ensure proper provision for the welfare of animals used in tests to assess the biocompatibility of biomaterials. It was last revised in July 2006 and is centred upon the concept of the three Rs: reduction, refinement and replacement. It recommends reductions in the overall number of animals used, refinement of test methods and replacement of animal tests with other scientifically valid means. Overall, the standards promote maximizing the use of scientifically sound non-animal tests [59]. In vitro immune response analysis not only offers the potential to fulfil these three Rs, but also substantial reductions in experimental time and resources.

Ideally, techniques of in vitro immune evaluation would be standardised to allow for direct comparisons across materials. Standardisation of technique would require incorporation into the International Organisation of Standardisation (ISO) standards for biological evaluation of medical devices (ISO10993), which remains the basis of FDA regulatory processes which biomaterials must pass to be cleared for clinical use [10,60]. However, as biomaterials are currently produced from both commercial and educational institutions, worldwide standardisation is unlikely to occur due to both the time and financial restrictions, and often an ill desire for materials to be directly compared [61,62].

In vitro assays of immunogenicity to date have evaluated the viability, maturation and activation of *innate* immune cells, with the most studies focusing on activation as it potentially provides the more relevant predictors of immune tolerance for researchers. Elements of the M1/M2 foreign body response can be evaluated to predict immune response early and determine likelihood of acceptance, which would negotiate undesirable local or systemic responses at the clinical level. It has been shown in vivo that a higher M2:M1 ratio is associated with positive healing outcomes [28,29]. In vitro immune response testing of biomaterials using M1/M2 assays may better provide in vitro/in vivo correlation. A good example of this is from Wolf et al. (2014), who combined in vitro human PBMC protein secretion profiles from CBA analysis, with quasi-mechanistic in silico analysis to produce a good correlation between in vitro evaluation and in vivo acceptance [63]. Moreover, Jannasch et al. (2017) proposed an in vitro testing platform, targeted as an alternative to in vivo animal studies. Their platform involved evaluation of material-induced chemotaxis and tissue-remodelling by monocyte-derived primary macrophages, and found a high correlation between their in vitro and state-of-the-art in vivo study data after pre-treatment of their testing surfaces with human blood plasma. Significant differences in material dependency were shown with titanium exhibiting a lower tissue remodelling capacity compared to polytetrafluorethylene, silicone and polyethylene [64,65]. This high vitro/in vivo correlation confirms that in vitro data alone can be a credible tool for informing decision making in biomaterial development, whilst also being time and cost efficient. Interestingly, human blood plasma treatment has been shown to downregulate proinflammatory cytokine expression in two recent studies evaluating calcium phosphate discs and titanium dental implants [66,67].

It is clear that assessing immunogenicity in vitro, without a complete inflammatory environment, will never be fully sensitive. Acquired immune responses can take months to develop, and in vitro studies have thus far been limited to cell viability assessment of T and B lymphocytes exposed to electrospun polydioxanone–elastin biomaterial blends [20,21]. Therefore, this may be best studied using in vivo models to enhance accuracy. The main aim of any proposed in vitro immune analysis should, therefore, be to capture short-term, highly reactive biomaterials. This would then prevent unnecessary and unethical future in vivo investigations. Based on the literature presented here, the simplest way to do this would likely be to study M1/M2 markers of pro- and anti-inflammation.

Overall, this review highlights the necessary role of in vitro immune response assessment for novel biomaterials. Immune response-focused in vitro data would provide pre-clinical indications of in vivo acceptance to researchers and clinicians, which can then be used to make informed decisions about whether a material should move to animal and human studies. With uptake of in vitro immune response analysis, not only substantial reductions in experimental time and resources, including unnecessary and unethical animal use, but also a desirable decrease in the number of inappropriate biomaterials reaching clinic, is possible. This improvement in bench to bedside safety is paramount to reduce patient harm.

## Figures and Tables

**Figure 1 jfb-10-00031-f001:**
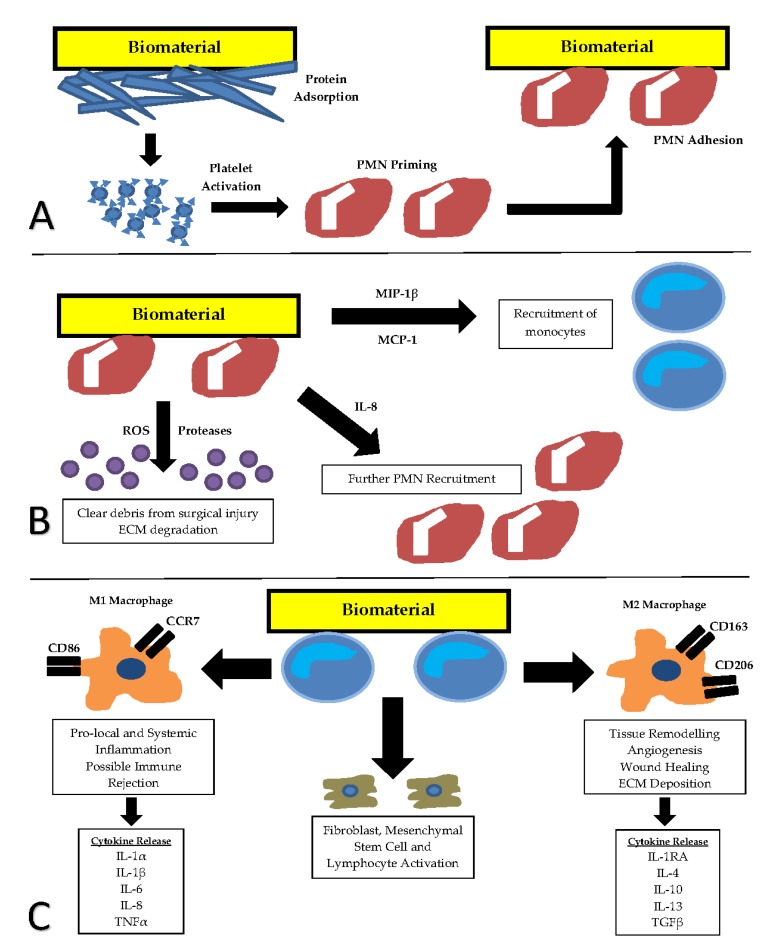
The sequence ofthe foreign body response to implanted biomaterials. (**A**) Adsorption of blood proteins and platelet activation results in priming and activation of polymorphonuclear (PMN) leukocytes. (**B**) Acute inflammation is defined by PMN presence at the site of repair. Monocytes are stimulated in the transition from acute to chronic inflammation. (**C**) Chronic inflammation is defined by mononuclear leukocyte presence. Cell surface marker production and inflammatory cytokine expression from two functional macrophage sub-types M1 (classically activated) and M2 (alternatively activated) form the basis for markers of immune evaluation. CCR7—C-C chemokine receptor type 7, CD86—cluster of differentiation 86, CD 163—cluster of differentiation 163, CD206—cluster of differentiation 206, ECM—extracellular matrix, IL-1α—Interleukin-1 alpha, IL-1β—Interleukin-1 beta, IL-1RA—Interleukin-1 receptor antagonist, IL-4—interleukin-4, IL-6—interleukin-6, IL-8—interleukin-8, IL-10—interleukin-10, IL-13—interleukin-13, MIP-1β—macrophage inflammatory protein-1 beta, MCP-1—monocyte chemoattractant protein-1, PMN—polymorphonuclear leukocyte, ROS—reactive oxygen species, TGFβ—transforming growth factor beta, TNFα—tumour necrosis factor alpha.

**Figure 2 jfb-10-00031-f002:**
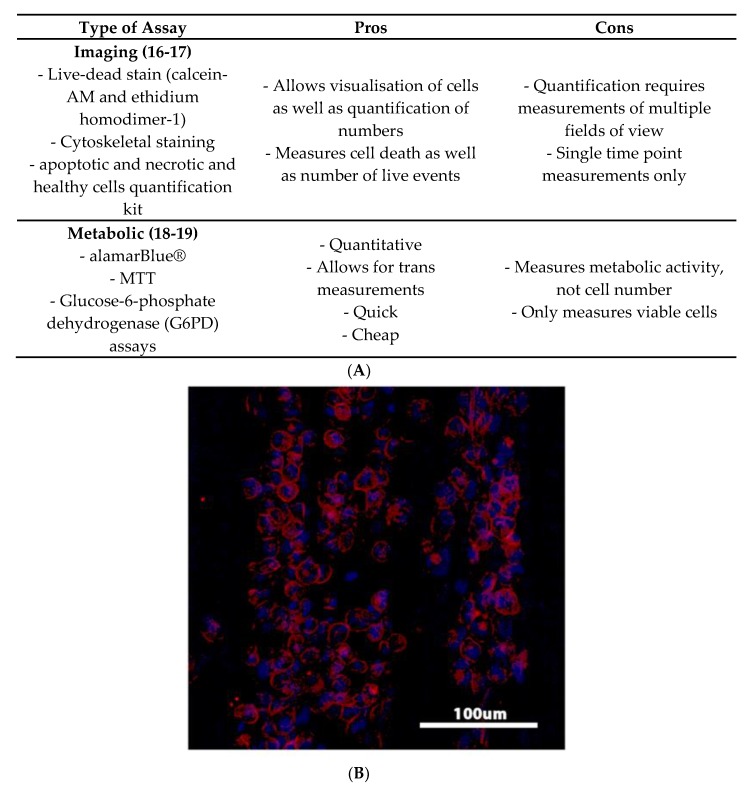
(**A**) Table discussing pros and cons of various measures of viability. (**B**) Example of macrophage cells exposed to a decellularised ECM and stained with phalloidin for cytoskeletal visualisation, and counter stained with DAPI to visualise the nuclei. DAPI = 4′,6-diamidino-2-phenylindole

**Table 1 jfb-10-00031-t001:** Summary of common cytokines used to assess likely immune rejection or acceptance of a material.

Pro-Inflammatory Cytokines Assessed Associated with Immune Rejection	Anti-Inflammatory Cytokines Assessed Associated with Immune Acceptance
IL-1α	IL-1RA
IL-1β	IL-4
IL-6	IL-10
IL-8	IL-13
IL-17A	TGFβ
CXCL10	
TNFα

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
