# Peer review of "The Role of In Vitro Immune Response Assessment for Biomaterials"

_jfb, 2019, doi:10.3390/jfb10030031_

Round 1
Reviewer 1 Report
This review provides a nice overview of the immune assays that can be found in the literature and establishes the importance of these assays for translation of new biomaterials to the clinic. Section 4 mention the usual cell types evaluated including peripheral blood mononuclear cells, several macrophage types, dentritic cells and T and B lymphocytes. Studies revised include cell viability, cell maturation and activation studies and finally elisa assays, protein microarrays and gene expression. While cell viability/maturation/ activation are simply descriptive sections. Elisa, microarrays and gene expression sections include some discussion indicating the shortcoming of the techniques and some contradictory reported results. And while RT-PCR is mentioned this technique is rather being substituted by q-PCR. This is suggested somehow but no clearly established as it should be for the interest of a new generation of readers.
New approaches with a greater potential are only mentioned in lines 353-363. It would be desirable to expand this section with the latest development including an opinion from the authors about new or missing approaches. Also , only 16 out of the 59 references are from the last 5 years. I would suggest an update of the references for expanding section 5. The subject of the review is really important, a hot topic of research and deserves this update.
Author Response
Reviewer 1
This review provides a nice overview of the immune assays that can be found in the literature and establishes the importance of these assays for translation of new biomaterials to the clinic.
Section 4 mention the usual cell types evaluated including peripheral blood mononuclear cells, several macrophage types, dentritic cells and T and B lymphocytes. Studies revised include cell viability, cell maturation and activation studies and finally elisa assays, protein microarrays and gene expression. While cell viability/maturation/ activation are simply descriptive sections. Elisa, microarrays and gene expression sections include some discussion indicating the shortcoming of the techniques and some contradictory reported results. And while RT-PCR is mentioned this technique is rather being substituted by q-PCR. This is suggested somehow but no clearly established as it should be for the interest of a new generation of readers.
We thank the reviewer for their comments, we felt it was important to include some of the older studies that utilised RT-PCR, but the reviewer is correct that the relationship between RT-PCR and more modern studies may not have been clear. We have now highlighted and made clear that this method is outdated and has been replaced by qPCR.
New approaches with a greater potential are only mentioned in lines 353-363. It would be desirable to expand this section with the latest development including an opinion from the authors about new or missing approaches. Also , only 16 out of the 59 references are from the last 5 years. I would suggest an update of the references for expanding section 5. The subject of the review is really important, a hot topic of research and deserves this update
We thank the reviewer for this comment and have used this suggestion to both improve the manuscript, as suggested and added a number of more up-to-date references.
Reviewer 2 Report
Summary: Authors introduce the relevance of the study of the immune response for novel synthetic materials to be used for grafting. In vitro tests for the assesment of the immune response are presented and discussed.
Overall opinion: the topic is of interest to the audience of the journal and provides an interesting summary of available in vitro techniques to evaluate the immune response. However, major revisions are needed to provide more attractive information to the reader. Namely, the absence of figures and tables in the manuscript is a clear flaw of the manuscript.
Specific points:
-Section 3: please provide a figure showing the sequence of the foreign body response to implanted biomaterials.
-Section 4.1: please provide a table with the most common stains (also showing commercial names and suppliers) used to evaluate cell viability. Also provide a picture of a stained sample to get a first insight of the outcome of the test.
-Sections 4.2 to 4.5: please provide a table with the most common markers, commercial kits and suppliers used for these studies. Any figure from the examples described in the text will be also highly appreciated.
Author Response
Reviewer 2
Summary: Authors introduce the relevance of the study of the immune response for novel synthetic materials to be used for grafting. In vitro tests for the assesment of the immune response are presented and discussed.
Overall opinion: the topic is of interest to the audience of the journal and provides an interesting summary of available in vitro techniques to evaluate the immune response. However, major revisions are needed to provide more attractive information to the reader. Namely, the absence of figures and tables in the manuscript is a clear flaw of the manuscript.
Thank you for your comments and suggestions. We have now added the three suggested tables and figures, which we can see will make the work much more digestible to the readers.
Specific points:
-Section 3: please provide a figure showing the sequence of the foreign body response to implanted biomaterials.
Diagram has been added as suggested
-Section 4.1: please provide a table with the most common stains (also showing commercial names and suppliers) used to evaluate cell viability. Also provide a picture of a stained sample to get a first insight of the outcome of the test.
We have added a table highlighting the main methods used to assess cell viability, and the relative pros and cons of each. We did not include supplier details as there are many, and these would be available in the original articles which we have cited. We also included a diagram of macrophages cultured on a decellularised ECM, from our lab, as we felt this avoided any copyright issues but still demonstrated the type of images these techniques would provide.
-Sections 4.2 to 4.5: please provide a table with the most common markers, commercial kits and suppliers used for these studies. Any figure from the examples described in the text will be also highly appreciated.
We have added a table summarising the main cytokine used to assess immune rejection or immune acceptance. We have not included suppliers as there are many different methods used to assess the levels of these and each have a number of different suppliers. We also did not include a figure from the cited literature as this would likely be in the form of a graph comparing levels, which out of context in this review may not mean as much to the readers.
Reviewer 3 Report
The review paper was well organized and written. The authors clearly addressed the importance of in-vitro immune biocompatibility test methods in the manuscript. Current in-vitro immune response assessment for biomaterials was well addressed as well. The organization of the manuscript was also good, which made it easy to understand and follow. The reviewer recommends it to be accepted at its present form.
Author Response
We thank the reviewer for their kind comments
Round 2
Reviewer 2 Report
Authors have addressed all my concerns. In my opinion, it can be accepted in its current format.